# Quality and Diversity Optimization Even From Offline Homogeneous Dataset

## Abstract

We investigate the challenge of promoting diversity in offline reinforcement learning (RL), where agents must develop diverse strategies despite being trained on homogeneous datasets with limited behavioral variation. Existing offline RL approaches, including those leveraging expectation-maximization algorithms for unsupervised clustering, often struggle with either insufficient diversity or performance degradation in such settings. To overcome these limitations, we introduce a novel Unique Behavior objective function that can be *directly computed to quantify the distinctiveness between agents*, eliminating the need for additional estimators and reducing potential estimation errors. By maximizing uniqueness, our approach encourages agents to learn diverse behaviors effectively, even when the training data lacks variety. Extensive experiments on D4RL MuJoCo and Atari benchmarks demonstrate that our method achieves significant behavioral diversity while maintaining strong performance, even from homogeneous training data.

## 1 Introduction

Learning to perform tasks with a range of behaviors, often referred to as quality-diversity (QD) optimization, is an increasingly prominent topic in reinforcement learning (RL) (Fontaine and Nikolaidis, 2021; Cully and Demiris, 2017; Pugh et al., 2016). Existing approaches for promoting diverse behaviors have primarily focused on online RL settings (Nilsson and Cully, 2021; Pierrot et al., 2022), where diversity improves exploration (Hong et al., 2018) and facilitates unsupervised skill discovery (Eysenbach et al., 2019; Sharma et al., 2020; Chen et al., 2024). These methods ultimately enhance policy robustness and adaptability. However, encouraging such diversity in offline RL (Kostrikov et al., 2021; Wu et al., 2020; Kumar et al., 2019; Fujimoto and Gu, 2021; Wang et al., 2020b) remains relatively underexplored and inherently challenging due to the static nature of training datasets, which restricts access to new experiences.

Learning diverse behaviors in offline RL is just as important as in online settings. Real-world tasks often admit multiple valid solutions, where each behavior may represent a different strategy, preference, or trade-off. Recent works such as SORL (Mao et al., 2024) and DIVEOff (Osa and Harada, 2024) aim to capture this solution diversity by extracting varied behaviors from heterogeneous datasets. SORL relies on expectation-maximization (EM) to cluster demonstrations and train separate policies per cluster, while DIVEOFF incorporates mutual information (MI) into EM to enhance diversity. Since MI is generally intractable due to the unavailability of state visitation distributions (Osa et al., 2022; Li et al., 2017; Laskin et al., 2022; Li et al., 2021; Osa and Harada, 2024), DIVEOff approximates MI using variational estimators, which may introduce additional error and instability.

In this work, we reinterpret the MI-based formulation of path diversity and decompose it into two components: *behavior diversity* and *state visitation diversity*. While full path diversity remains intractable, we show that behavior diversity is directly computable, whereas state visitation diversity is not. We provide a theoretical guarantee that behavior diversity forms a lower bound on full path diversity. Moreover, we show empirically that optimizing behavior diversity improves diversity at both the behavior and trajectory levels, while also stabilizing training. To this end, We propose the *Unique Behavior (UB)* objective, which directly optimizes behavior diversity via a mutual information-based reward. Unlike methods such as SORL and DIVEOFF that rely on behavior clustering, UB promotes diversity by explicitly maximizing action uniqueness across policies.

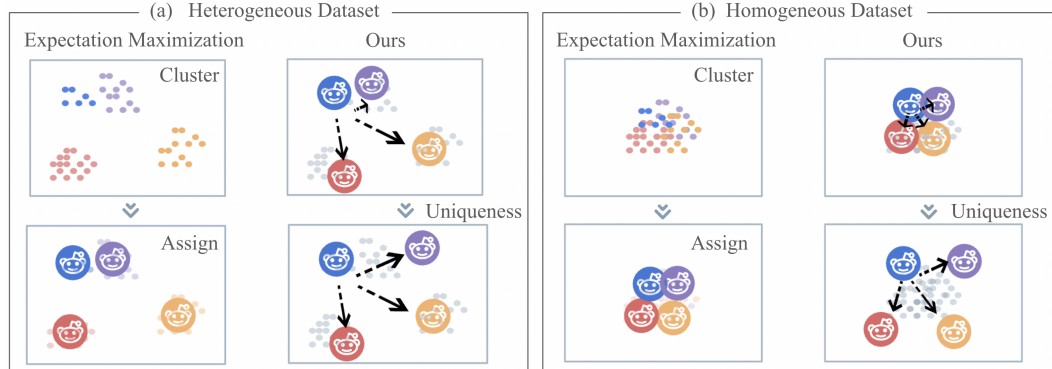

Figure 1: While EM-based methods can capture diverse behaviors in heterogeneous datasets (panel (a)), they struggle with homogeneous datasets lacking inherent variety (panel (b)). In contrast, our method promotes diversity by directly encouraging learned policies to differ in their action distributions. Dashed arrows indicate divergence between policy outputs, which underlies our uniqueness score and ensures behavioral distinctiveness even when the dataset lacks diversity.

This allows our method to discover multiple distinct solutions even from homogeneous datasets. In our context, we define **heterogeneity** as the presence of diverse user behaviors in the dataset, e.g., in driving scenarios, ranging from cautious followers to aggressive lane changers. Conversely, **homogeneity** reflects uniform behavioral patterns, where most data points exhibit similar decision-making. As shown in Figure 1, such homogeneity poses a challenge for EM-based methods, which fail to capture meaningful diversity in the absence of inherent behavioral variation. In contrast, Figure 2 (see demo GIF at https://1drv.ms/i/c/f80c15fe86484a40/EVD5RmHH8uxJrUlVMRzd-YUBMfCLak7YmzuuzhBlYWHrLg?e=7Uk4nM) showcases 4 consecutive timesteps from three distinct agents trained under the same task, demonstrating that our method is able to learn diverse behaviors even from a homogeneous dataset. *Note: For improved clarity, we added visual symbols such as arrows and circles to the GIF.*

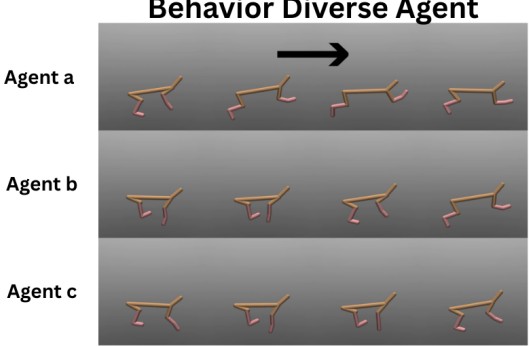

Figure 2: Visualization of behavioral diversity across agents on the same task. Each row corresponds to a different trained agent performing the task trained on the HalfCheetah-Medium-v2 dataset. All agents are shown at the same 4 consecutive timesteps (selected with fixed seed 52), revealing varied locomotion styles and state trajectories despite identical environment and initialization.

While the UB objective promotes behavioral diversity, our ultimate goal is to ensure these behaviors are useful and high-performing. Diverse policies may select out-of-distribution (OOD) actions, leading to unreliable value estimates. To mitigate this, we adopt a conservative Q-learning strategy inspired by Kumar et al. (2020), which uses an ensemble of critics and selects the minimum Q-value across the ensemble. This penalizes uncertain or unsupported actions and helps ensure that the diverse behaviors generated by UB remain grounded in reliable, high-quality decision-making.

We evaluate our method on the standard D4RL benchmark (Fu et al., 2020) and its diverse extension (Osa and Harada, 2024), covering both heterogeneous and homogeneous settings. Our results demonstrate that the proposed approach achieves a favorable balance between performance and diversity, consistently discovering high-quality, distinct behaviors across tasks. These findings demonstrate that our method provides a principled approach to promoting policy diversity in offline RL. By encouraging distinct behaviors, it enables agents to generalize beyond the limited data distribution—even in constrained, static training environments.

## 2 RELATED WORK

### 2.1 MUTUAL INFORMATION (MI) IN RL

Mutual Information (MI) has been widely used in both online and offline RL as a tool to capture statistical dependencies between states, actions, and representations. In what follows, we first review how MI has been applied in online RL for exploration and representation learning, and then discuss its emerging role in offline RL settings.

In online RL contexts, MI has been primarily utilized to encourage agents to explore diverse skills through empowerment (Leibfried et al., 2019; Mohamed and Jimenez Rezende, 2015; Klyubin et al., 2005; 2008) or information-theoretic curiosity (Still and Precup, 2012; Bai et al., 2021; Tao et al., 2020), measuring dependencies between successive states to promote diverse interactions within the environment. Beyond exploration, MI has also been instrumental in advancing representation learning (Anand et al., 2019; Stooke et al., 2021; Nachum et al., 2019; Schwarzer et al., 2021; Mazoure et al., 2020), where it quantifies dependencies between states and their representations or between state-action pairs and their corresponding representations. Additionally, MI has been applied in multi-agent reinforcement learning (MARL) to enhance coordination (Jaques et al., 2019; Konan et al., 2022) and promote diversity among agents (Jiang and Lu, 2021; Li et al., 2021; Liu et al., 2022; Wang et al., 2020a).

In offline RL, MI has seen limited application, though recent works have begun to explore its potential to address the distribution shift that arises when out-of-distribution actions are queried. Lou et al. (2022) proposed an action embedding model based on MI, helping the value function generalize to OOD actions that may still yield high rewards. Ma et al. (2024) proposed using **M**utual **I**nformation between **S**tates and **A**ctions (MISA) as a regularizer to encourage policy expressiveness in offline reinforcement learning; by encouraging high MI, the policy is constrained to remain closer to the behavior policy and the data manifold.

The above approaches in online and offline RL ultimately optimize for a single solution, without addressing both diversity and quality within a task. Our work highlights this gap and shows that MI can serve a fundamentally different purpose: directly optimizing for solution diversity and performance in offline RL. This perspective sets the stage for recent advances in Quality-Diversity (QD) optimization, which aim to uncover a spectrum of high-performing yet distinct behaviors from offline data — a challenge we now turn to.

### 2.2 QUALITY AND DIVERSITY OPTIMIZATION IN OFFLINE RL

QD optimization has emerged as a promising direction in offline RL, seeking to uncover a range of high-performing policies that reflect different strategies, preferences, or trade-offs. Unlike conventional RL approaches that converge to a single optimal policy, QD methods aim to populate the solution space with diverse yet effective behaviors. This is particularly valuable in real-world applications, where solutions are rarely one-size-fits-all. Various strategies have been developed to capture behavioral diversity from datasets with contributions from multiple users, each potentially reflecting distinct styles and preferences. For example, CLUE (Liu et al., 2023) leverages a Variational Autoencoder (VAE) framework to encode state-action pairs into latent representations. By identifying different subsets of data as distinct behavioral targets, CLUE assigns intrinsic rewards based on latent distance from other data points, thus facilitating the learning of diverse behaviors.

Other prominent approaches such as SORL (Mao et al., 2024) and DIVEOFF (Osa and Harada, 2024) utilize expectation-maximization (EM) techniques to capture diversity. Specifically, SORL clusters behaviors using an EM algorithm, updating latent behavior policies iteratively to maximize a

lower bound of the posterior log-likelihood. DIVEOFF further extends this concept by incorporating MI into the EM procedure, employing a VAE to condition policies on latent variables. Given the intrinsic intractability of MI due to unknown state visitation distributions (Osa et al., 2022; Li et al., 2017; Laskin et al., 2022; Li et al., 2021; Osa and Harada, 2024), DIVEOFF resorts to variational approximations, potentially leading to approximation errors and training instability.

Several limitations exist in the aforementioned approaches. For example, the SORL framework explicitly states that it "not only optimizes the performance of the policies but also preserves the inherent behavioral diversity found in heterogeneous datasets", indicating its reliance on naturally diverse data. Similarly, CLUE assumes access to a large offline dataset accompanied by expert trajectories, which may not hold in practice. Moreover, methods such as DIVEOFF have only been evaluated on datasets with heterogeneous behavior, leaving their effectiveness on homogeneous datasets unexamined.

Collectively, these approaches under-address a critical challenge: how to ensure policy diversity when the offline dataset is homogeneous and lacks inherent behavioral variation. Addressing this limitation, our work introduces an alternative formulation that explicitly promotes diversity within the action space while preserving high task performance. By quantifying action-level uniqueness across agents, our method incentivizes the discovery of distinct behaviors even in the absence of diverse demonstrations.

## 3 PRELIMINARIES

**Reinforcement Learning**

A Markov Decision Process (MDP) is defined by the tuple $M = (S, A, P, r, \rho, \gamma)$. Here, $S$ and $A$ are the state and action spaces respectively. The reward function $r(s, a)$, with state $s$ and action $a$, has a range of $[-r_{\max}, r_{\max}]$. The transition function is represented by $P(s'|s, a)$, $\rho$ is the initial state distribution, and $\gamma \in (0, 1)$ denotes the discount factor. We consider Markovian policies, $\pi \in \Pi$, which map states to distributions over actions. The value function $V^\pi(s) = \mathbb{E}_{a_t \sim \pi, s_t \sim P} \left[ \sum_{t=0}^{\infty} \gamma^t r(s_t, a_t) \right]$ calculates the expected discounted return from any initial state. This leads to the overall expected return: $\eta(\pi) = \sum_{s \in S} \rho(s) V^\pi(s)$. The state-action value function is then defined as: $Q^\pi(s, a) = \mathbb{E}_{a_t \sim \pi, s_t \sim P} \left[ r(s, a) + \sum_{t=1}^{\infty} \gamma^t r(s_t, a_t) \mid s_0 = s, a_0 = a \right]$.

## 4 DIVERSE SOLUTIONS IN OFFLINE RL

Encouraging diverse behaviors in RL has become increasingly important for promoting robustness, enabling personalization, and uncovering multiple valid solutions to the same task. However, in the offline RL setting, quantifying and optimizing behavioral diversity poses significant challenges due to the lack of environment interaction and limited access to agent-specific trajectory statistics.

In this section, we present two complementary formulations for measuring policy diversity: *path diversity*, which considers the entire sequence of states and actions, and *behavior diversity*, which focuses on variations in action selection under shared states. While path diversity provides a rich theoretical foundation, it is generally intractable, especially in offline RL. To address this, we introduce behavior diversity as a tractable surrogate and the basis for our proposed optimization framework.

### 4.1 INTRACTABLE PATH DIVERSITY

We begin by formalizing *path diversity*, which captures how distinct the trajectories generated by different policies are. For each agent $\pi^i$, we define the *path uniqueness* of its trajectory $\tau = (s_0, a_0, \ldots, s_T, a_T)$ based on the mutual information between the trajectory $\mathcal{T}$ and the policy identity $\Pi$:

$$I(\mathcal{T}; \Pi) = \mathbb{E}_{(\pi, \tau) \sim p(\Pi, \mathcal{T})} \left[ \log \frac{p(\tau | \pi)}{p(\tau)} \right]. \tag{1}$$

We define the *path uniqueness* for policy $\pi^i$ as:

$$U_{\text{Path}}^{\pi^i}(\tau) = \log \frac{p(\tau | \pi)}{p(\tau)} \tag{2}$$

Equation 2 captures trajectory visitation uniqueness. Since each policy may generate a distinct trajectory, a high path uniqueness for each of these trajectories implies that the policies are behaving in meaningfully different ways. Consequently, when individual path uniqueness scores are high, the overall population exhibits a high degree of path diversity.

While theoretically appealing, MI is intractable and difficult to compute. This intractability arises from the need to model high-dimensional distributions over policies and trajectories. Estimating the joint distribution $p(\pi, \tau)$, as well as the marginals $p(\pi)$ and $p(\tau)$, requires access to density estimates in complex spaces, both of which are computationally prohibitive. As a result, most prior works adopt variational approximations (Li et al., 2021), contrastive learning (Laskin et al., 2022), or discriminator (Osa et al., 2022; Li et al., 2017) to quantify or encourage diversity.

These approximation strategies are already nontrivial in online RL. In offline settings, where no additional interaction with the environment is allowed, the problem becomes even more severe. In the following, we show how our proposed approach avoids these intractable components while still enabling effective diversity optimization.

## 4.2 A Tractable Surrogate: Behavior Diversity

In real-world settings (e.g., partially observable MDPs, contextual MDPs), the states an agent visits can depend on its complete history and hence on $\pi$ itself. To capture this, we explicitly allow each time-step distribution to vary, writing $p(\tau \mid \pi) = \prod_{t=0}^{T} p^t(s_t \mid \pi) \cdot p(a_t \mid s_t, \pi)$, and $p(\tau) = \prod_{t=0}^{T} p^t(s_t) \cdot p(a_t \mid s_t)$. Note that each $p^t(\cdot)$ can be different at every time step; this motivates the notation $p^t$ used throughout Appendix A.3.

We rewrite the *path uniqueness* for policy $\pi^i$ as:

$$U_{\text{Path}}^{\pi^i}(\tau) = \sum_{t=0}^{T} \log \frac{p(a_t|s_t, \pi^i)}{p(a_t|s_t)} + \sum_{t=0}^{T} \log \frac{p^t(s_t|\pi^i)}{p^t(s_t)}. \tag{3}$$

The first term captures action-level distinctiveness, and the second term captures state-level visitation uniqueness. Equation 3 reveals that only the second term is intractable, whereas the first term is directly computable from policy output probabilities. We therefore focus on *behavior diversity*, which considers only the distinctiveness of action selection under the same state across different policies. This behavior diversity corresponds to the mutual information between actions $\mathcal{A}$ and policies $\Pi$, conditioned on states $\mathcal{S}$: $I(\mathcal{A}; \Pi \mid \mathcal{S}) = \mathbb{E}_{(a,\pi) \sim p(\mathcal{A}, \Pi | \mathcal{S})} \left[ \log \frac{p(a_t|s_t, \pi)}{p(a_t|s_t)} \right]$.

For each policy $\pi^i$, we define the *behavior uniqueness* of a trajectory $\tau$ as: $U_{\text{Behavior}}^{\pi^i}(\tau) = \sum_{t=0}^{T} \log \frac{p(a_t|s_t, \pi^i)}{p(a_t|s_t)}$. Unlike path diversity, this formulation is tractable in offline settings. The policy-specific term $p(a_t \mid s_t, \pi^i)$ is available from the policy network, and the marginal can be computed as: $p(a_t \mid s_t) = \sum_{i=1}^{M} p(a_t \mid s_t, \pi^i) \cdot p(\pi^i)$, where $p(\pi^i) = \frac{1}{M}$ under a uniform prior, *without loss of generality*, to reflect our objective of discovering diverse solutions of equal importance. That is, each distinct policy $\pi^i$ is treated as an equally valuable solution to the task, without imposing preference or weighting among them. Although $U_{\text{Path}}^{\pi^i}$ is not directly computable, it can be shown that behavior uniqueness serves as a lower bound:

$$U_{\text{Path}}^{\pi_\theta^i}(\tau) = \sum_{t=0}^{T} \log \frac{p(a_t \mid s_t, \pi^i)}{p(a_t \mid s_t)} + \sum_{t=0}^{T} \log \frac{p^t(s_t \mid \pi^i)}{p^t(s_t)} \geq U_{\text{Behavior}}^{\pi_\theta^i}(\tau). \tag{4}$$

Adapting the mutual information concept from Equation 1 to the context between the state $\mathcal{S}$ and the policy $\Pi$, we have: $I(\mathcal{S}; \Pi) = \mathbb{E}_{(\pi,s) \sim p(\Pi, \mathcal{S})} \left[ \log \frac{p(s|\pi)}{p(s)} \right]$. Given the non-negative nature of mutual information, this formulation supports the validity of Inequality 4. Thus, *maximizing behavior uniqueness promotes a lower bound on overall path-level uniqueness.*

Let $\pi_\theta^i$ be a differentiable policy parameterized by $\theta$, and $\eta(\pi_\theta^i)$ denote its expected return. We define a diversity-augmented objective as:

$$\mathcal{L}(\pi_\theta^i) = -\eta(\pi_\theta^i) - \lambda \cdot U_{\text{Behavior}}^{\pi_\theta^i}, \tag{5}$$

where $\lambda \geq 0$ balances task performance and behavior uniqueness.

### 4.3 INTUITION AND PRACTICAL ALGORITHM

In Figure 3, we use a 2D maze environment to illustrate how increasing behavior diversity can lead to greater path diversity. In this environment, the corridors are shown in white and the walls in gray. The goal is for agents to navigate from various starting positions to a target located at the bottom-left corner. Agents only receive rewards when they are within a 0.5-unit radius of the goal, which is visually indicated by a circle. The dataset consists of segments $(s, a, s')$, where $s$ is the current state, $a$ is the action taken, and $s'$ is the subsequent state. By maximizing behavior diversity, the experiment showed that agents followed different routes to reach the destination. This outcome is expected, as a trajectory is a sequence of states and actions, and small differences in local states accumulate over the length of the trajectory, leading to distinct paths.

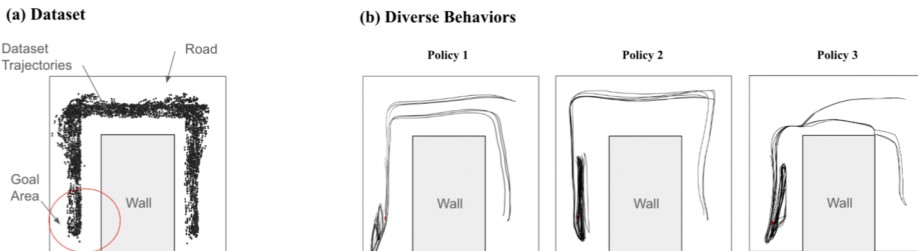

Figure 3: We demonstrate that increasing behavior diversity can lead to greater path diversity using a 2D maze environment. (a) The state-action pairs in the offline dataset. (b) *The variation in paths executed by different policies underscores the effectiveness of our approach in optimizing behavior diversity and path diversity from the same training dataset.*

We build our method on top of the stochastic policy optimization technique, EDAC (An et al., 2021), which is enhanced by $N$ ensemble Q-function networks. This ensemble helps reduce the overestimation of values for actions not present in the dataset, ensuring performance is maintained while pursuing diversity. The policy objective function in EDAC mirrors that of soft-actor-critic (SAC) (Haarnoja et al., 2018), and is formulated as:

$$D_{\text{KL}} \left( \pi_\theta(\cdot|s) \middle\| \frac{\exp(Q_w^\pi(s, \cdot))}{Z^\pi(s)} \right), \tag{6}$$

where $Z^\pi(s)$ normalizes the distribution. To incorporate diversity into the policy objective, we employ a reward shaping technique (Hu et al., 2020), by redefining the reward of each agent $\pi_\theta^m$ as $r'_m(s_i, a_i) = r(s_i, a_i) + \lambda U_{\text{Behavior}}^{\pi_\theta^m}(a'_{i,m}|s'_i)$, where $U_{\text{Behavior}}^{\pi_\theta^m}(a'_{i,m}|s'_i) = \log \frac{p(a'_{i,m}|s'_i, \pi_\theta^i)}{p(a'_{i,m}|s'_i)}$ represents the uniqueness of the action $a'_{i,m}$ chosen by policy $\pi_\theta^m$ at next state $s'_i$. This modification encourages policies to maximize both expected returns and behavior diversity.

Algorithm 1 in Appendix A.5 provides the implementation details of our approach. We adopt the same hyperparameters as the EDAC framework (An et al., 2021), with an additional hyperparameter, $\lambda$, which balances task performance and policy diversity. This $\lambda$ was set to 1.0 across all offline datasets, as it generally provides a good trade-off between these two objectives. In the medium-expert Walker2d environment from the diverse D4RL dataset, we reduced $\lambda$ to 0.5 to prevent over-diversification, which can hinder performance when high-reward trajectories dominate the dataset. Please refer to Appendix A.2 for detailed parameter settings.

## 5 RESULTS AND EVALUATION

### 5.1 COMPARISON WITH BASELINE METHODS

We conducted experiments using both the standard (Fu et al., 2020) and diversified versions (Osa and Harada, 2024) of the D4RL datasets to evaluate our method. Standard D4RL dataset is generated using a Soft Actor-Critic (SAC) policy, which typically results in homogeneous behavior patterns. In contrast, the Diverse D4RL Dataset is collected using a latent-conditioned policy that is specifically designed to capture a wider range of behaviors, making it more heterogeneous. The diverse D4RL

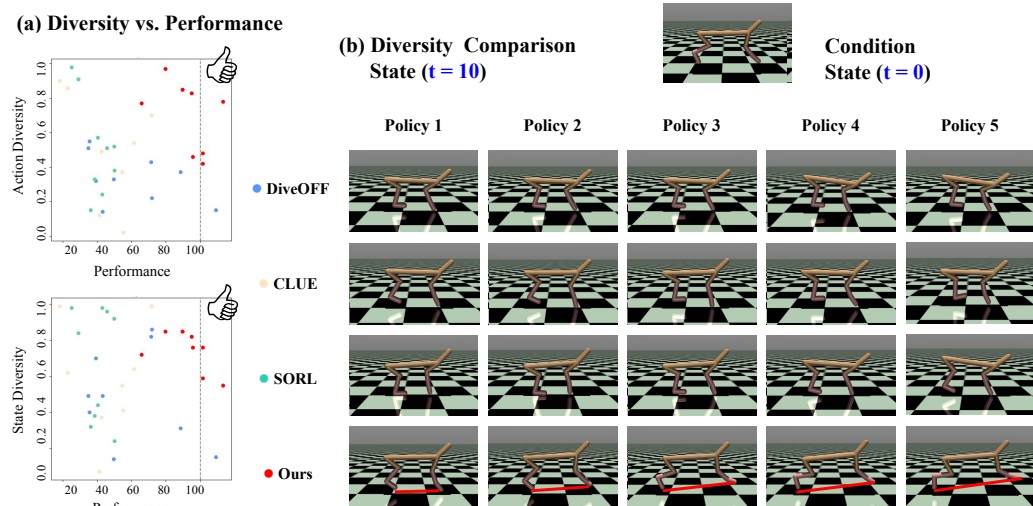

Figure 4: (a) The scatter plot illustrates the relationship between performance and diversity, in terms of action and state, across various methods and environments in the standard D4RL dataset. Each point represents a policy trained on a specific dataset, with different colors corresponding to different methods. Notably, **policies trained using our approach tend to cluster in the upper-right corner of the plot**, reflecting both high performance and diversity. (b) To visually compare the diversity between policies trained with baseline methods and our approach, we initialized the policies from a consistent initial state at $t = 0$ and rendered the resulting states after 10 steps. For additional visual comparisons, please refer to Figure 7 in Appendix A.6.

dataset exhibits greater diversity compared to the standard D4RL dataset due to the inclusion of multiple latent-conditioned policies.

We assessed the effectiveness of our approach by comparing it with several established baseline methods designed to meet performance and diversity criteria, including DIVEOFF (Osa and Harada, 2024), CLUE (Liu et al., 2023), and SORL (Mao et al., 2024). To ensure a fair comparison, we carefully replicated the results of the baseline methods using the source codes provided by the authors [1][2][3]. Notably, as the original SORL implementation was designed for discrete action spaces, we adapted its neural network architecture to fit our continuous action space framework.

We quantified the diversity of an agent's behavior using the metric proposed in (Osa and Harada, 2024; Parker-Holder et al., 2020): $D_{\text{div}} = \det\left(K\left(\phi(\pi_i), \phi(\pi_j)\right)_{i,j=1}^{M}\right)$, where $\phi(\pi) \in \mathbb{R}^l$ is the behavioral embedding of policy $\pi$, and $K : \mathbb{R}^l \times \mathbb{R}^l \to \mathbb{R}$ is a kernel function. Specifically, we used the squared-exponential kernel function: $k(\phi(\pi_i), \phi(\pi_j)) = \exp\left(-\frac{\|\phi(\pi_i) - \phi(\pi_j)\|^2}{2h^2}\right)$, with $\phi_s(\pi_i) = \mathbb{E}_{s \sim \pi_i, P}[s]$ to evaluate state diversity and $\phi_a(\pi_i) = \mathbb{E}_{s \sim \pi_i, P}[a]$ to assess action diversity.

Table 1 presents the experimental results, where we report both the performance and diversity scores averaged over five random seeds. For clarity, the highest-performing algorithms are highlighted in **bold**, while the highest diversity scores are underlined. Additionally, the diversity scores, where the corresponding performance was within one standard deviation of the top performance models, are also highlighted in **bold** to emphasize cases where high diversity does not come at the cost of performance. As shown, our approach outperformed all baseline models in terms of overall performance and diversity across the evaluated tasks. This was true for agents trained on both the standard and diversified D4RL datasets. The relationship between diversity and performance is further visualized in Figure 4, where our method consistently occupies the upper-right region, indicating a favorable trade-off. Notably, our method's performance remained competitive with single-agent algorithms that do not consider diversity (see Appendix A.4 and Table 3). These findings demonstrate that

---

[1] DIVEOFF: https://github.com/TakaOsa/DiveOff

[2] CLUE: https://openreview.net/forum?id=xJ7XL5Wt8iN

[3] SORL: https://github.com/cedesu/SORL/tree/main

Table 1: Performance and diversity metrics of our method compared to baselines, averaged over 5 seeds and 10 episodes per seed. We used $M = 5$ for standard D4RL and $M = 9$ for diverse D4RL. Performance is normalized as in Fu et al. (2020), and diversity is measured following Osa and Harada (2024).

| Datasets Type | | (a) Standard D4RL Dataset | | | | (b) Diverse D4RL Dataset | | | |
|---|---|---|---|---|---|---|---|---|---|
| Datasets | Metrics | **Ours** | **DIVEOFF** | **CLUE** | **SORL** | **Ours** | **DIVEOFF** | **CLUE** | **SORL** |
| Medium-Expert Hopper | Performance | **95.66±12.04** | 88.37±15.3 | 54.33±1.77 | 42.7±6 | **95.59±6** | 96.81±5.08 | 95.12±5.16 | 61.06±4.36 |
| | State Diversity | **0.76±0.11** | 0.31±0.3 | 0.55±0.29 | 0.98±0.02 | **0.98±0.02** | 0.1±0.08 | 0.93±0.06 | 0.95±0.03 |
| | Action Diversity | **0.46±0.22** | 0.37±0.42 | 0.37±0.21 | 0.24±0.19 | **0.66±0.26** | 0.2±0.01 | 0.09±0.08 | 0.9±0.09 |
| Medium-Expert Walker2d | Performance | **113.35±0.57** | 109±0.14 | 107.54±0.66 | 49.92±7.63 | **99.16±0.56** | 96.32±4.8 | 72.72±0.44 | 46.49±4.28 |
| | State Diversity | **0.55±0.04** | 0.15±0.11 | 0.45±0.18 | 0.92±0.06 | **0.9±0.1** | 0.6±0.37 | 0.99±0 | 0.97±0.04 |
| | Action Diversity | **0.78±0.17** | 0.15±0.14 | 0.15±0.07 | 0.52±0.18 | **0.93±0.03** | 0.25±0.33 | 0.79±0.07 | 0.89±0.07 |
| Medium-Expert Halfcheetah | Performance | **95.08±7.88** | 71.39±6.09 | 61.45±3.31 | 49.98±5.06 | **98.22±0.24** | 96.47±0.31 | 95.28±0.11 | **98.25±0.31** |
| | State Diversity | **0.82±0.34** | 0.82±0.2 | 0.64±0.37 | 0.24±0.18 | **0.98±0.03** | 0.46±0.41 | 0.71±0.26 | 0.38±0.23 |
| | Action Diversity | **0.83±0.16** | 0.43±0.35 | 0.54±0.35 | 0.38±0.11 | **0.99±0** | 0.38±0.24 | 0.74±0.14 | 0.74±0.27 |
| **Medium-Expert-Performance** | | **101.36** | 89.58 | 74.44 | 47.53 | **97.65** | 96.53 | 87.7 | 68.6 |
| **Medium-Expert-Diversity** | | **0.7** | 0.37 | 0.45 | 0.54 | **0.91** | 0.33 | 0.7 | 0.81 |
| Medium-Replay Hopper | Performance | **101.41±0.26** | 35.41±11.05 | 22.65±0.01 | 29.04±2.54 | **100.44±0.16** | 100.88±0.23 | 0.01±0.01 | 40.86±0.99 |
| | State Diversity | **0.76±0.17** | 0.4±0.21 | 0.62±0.26 | 0.84±0.12 | **0.24±0.15** | 0.02±0.03 | 0.94±0.11 | 0.77±0.07 |
| | Action Diversity | **0.42±0.25** | 0.55±0.17 | 0.86±0.18 | 0.91±0.03 | **0.27±0.14** | 0.01±0 | 0.99±0 | 0.76±0.19 |
| Medium-Replay Walker2d | Performance | **79.63±0.54** | 34.79±9.19 | 18.04±0.24 | 25±4.95 | **94.28±2.1** | 51.04±8.52 | 16.13±1.13 | 33.69±1.61 |
| | State Diversity | **0.85±0.13** | 0.49±0.37 | 0.99±0 | 0.98±0.03 | **0.52±0.15** | 0.59±0.47 | 0.99±0 | 0.95±0.04 |
| | Action Diversity | **0.97±0.03** | 0.51±0.01 | 0.9±0.15 | 0.98±0.02 | **0.72±0.14** | 0.42±0.51 | 0.99±0 | 0.92±0.04 |
| Medium-Replay Halfcheetah | Performance | **60.92±1.3** | 39.22±0.79 | 41.32±0.4 | 40.39±9.48 | **95.49±0.28** | 39.22±0.79 | 90.44±0.49 | 68.8±20.6 |
| | State Diversity | **0.94±0.03** | 0.7±0.4 | 0.07±0.04 | 0.44±0.3 | **0.99±0** | 0.48±0.44 | 0.95±0.06 | 0.16±0.06 |
| | Action Diversity | **0.98±0.02** | 0.32±0.15 | 0.12±0.01 | 0.57±0.3 | **0.99±0.01** | 0.32±0.15 | 0.98±0.02 | 0.19±0.07 |
| **Medium-Replay-Performance** | | **80.65** | 36.47 | 27.33 | 31.47 | **96.74** | 63.71 | 35.53 | 47.78 |
| **Medium-Replay-Diversity** | | **0.82** | 0.5 | 0.59 | 0.79 | 0.62 | 0.31 | **0.97** | 0.62 |
| Medium Hopper | Performance | **101.44±0.19** | 49.66±1.88 | 55.19±0.85 | 38.54±4.15 | **99.1±0.28** | 91.43±5.1 | 81.37±0.49 | 61.38±4.12 |
| | State Diversity | **0.59±0.31** | 0.14±0.1 | 0.41±0.15 | 0.82±0.23 | **0.95±0.07** | 0.19±0.28 | 0.99±0 | 0.96±0.03 |
| | Action Diversity | **0.48±0.31** | 0.33±0.1 | 0.02±0.01 | 0.33±0.24 | **0.47±0.13** | 0.34±0.45 | 0.55±0.62 | 0.9±0.07 |
| Medium Walker2d | Performance | **89.66±0.61** | 71.9±3.08 | 71.59±2.86 | 45.73±3.64 | 80.95±8.87 | **86.45±8.53** | 56.81±1.11 | 50.35±3.49 |
| | State Diversity | **0.85±0.15** | 0.86±0.1 | 0.99±0.01 | 0.96±0.03 | **0.99±0.01** | 0.68±0.36 | 0.99±0 | 0.96±0.03 |
| | Action Diversity | **0.85±0.19** | 0.22±0.23 | 0.7±0.11 | 0.51±0.26 | **0.99±0.01** | 0.59±0.36 | 0.91±0.08 | 0.85±0.23 |
| Medium Halfcheetah | Performance | **65.89±0.34** | 43.18±0.09 | 42.45±0.35 | 36.06±0.18 | 92.84±0.34 | 93.15±0.05 | 92.88±0.9 | **97.26±0.08** |
| | State Diversity | **0.72±0.21** | 0.49±0.48 | 0.73±0.11 | 0.32±0.22 | 0.98±0.01 | 0.23±0.44 | 0.75±0.2 | **0.2±0.19** |
| | Action Diversity | **0.77±0.13** | 0.14±0.01 | 0.49±0.09 | 0.15±0.08 | 0.98±0.01 | 0.45±0.3 | 0.91±0.05 | 0.27±0.23 |
| **Medium-Performance** | | **85.66** | 54.91 | 56.41 | 40.11 | **90.96** | 90.34 | 77.02 | 69.66 |
| **Medium-Diversity** | | **0.71** | 0.36 | 0.56 | 0.52 | **0.89** | 0.41 | 0.85 | 0.69 |

our method achieves an optimal balance between performance and diversity, offering a significant advantage in tasks that require both high efficiency and a wide range of behavioral strategies.

These results highlight the advantage of our approach: explicitly optimizing diversity via the Unique Behavior objective, without relying on clustering or dataset heterogeneity. This objective is both tractable and efficient, as it can be directly computed by averaging over all policies under a uniform prior. Such a formulation allows the discovery of multiple distinct and high-performing behaviors, even in behaviorally homogeneous datasets, where traditional clustering-based approaches may fail to identify meaningful diversity. Additional experiments on controllable diversity and on discrete-action Atari environments are provided in Appendix A.10 and Appendix A.11, respectively.

## 5.2 ABLATION STUDY

Our optimized diversity-and-performance algorithm for offline RL is formalized through the Unique Behavior objective function term (Equation 5). To evaluate its effectiveness, we conducted an ablation study on nine benchmark D4RL datasets, comparing models trained with our Unique Behavior term against those using a discriminator-based approximation, as adopted in prior work (Osa and

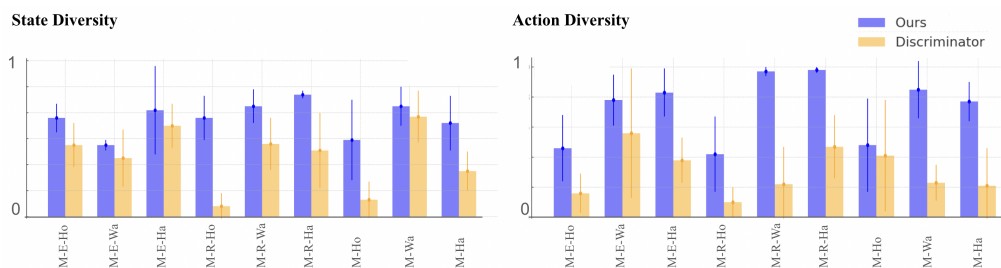

Figure 5: Comparison between our method and using a discriminator-based approximation across 9 datasets (3 tasks × 3 D4RL subsets; for example, M-E-Ho denotes the Medium-Expert subset of the Hopper task). The results show both state diversity (left) and action diversity (right), where the bars indicate mean values and the error bars represent standard deviations. As shown, our method consistently achieves higher diversity across all settings.

Harada, 2024; Osa et al., 2022; Li et al., 2017; Eysenbach et al., 2019). The results, averaged over five random seeds, are presented in Figure 5. They indicate that incorporating the Unique Behavior objective function maintains comparable performance metrics while significantly enhancing the diversity of agent behaviors. Notably, the benefit of the Unique Behavior objective function term is evident not only in the medium-replay and medium datasets, which naturally exhibit more diverse behaviors due to lower performance levels, but also in the medium-expert dataset. For further ablation studies on the impact of our UB objective and the sensitivity to hyperparameters (number of agents, number of critics and $\lambda$), please refer to Appendix A.7, Appendix A.8 and Appendix A.9.

## 5.3 ACTION DISTANCE FROM DATASETS

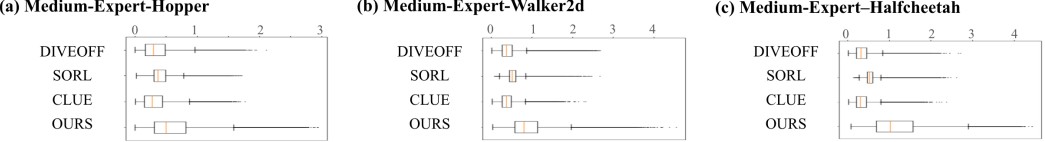

Figure 6: The Box and Whisker Plots depict the averaged distances between actions generated by policies and those sampled from the dataset under the same state. The distances were measured on the *medium-expert* level of the standard D4RL datasets. Each box represents the first quartile (Q1), the median, and the third quartile (Q3). The whiskers indicate 1.5 times the inter-quartile range (IQR), with outliers shown as individual points beyond the whiskers.

One of the key advantages of our method is its ability to learn actions outside the dataset distribution, enabling agents to exhibit diverse behaviors through intrinsic mechanisms. To quantitatively assess this benefit, we measured the average Euclidean distance between the actions selected by the agents and those typically found in the dataset. This metric is defined as: $E_{(s,a)\sim D, \hat{a}\sim\pi_\theta(\cdot|s)}[\|\hat{a} - a\|^2]$, where $D$ represents the dataset and $\pi_\theta(\cdot|s)$ denotes the actions chosen by the policy. We compared our approach with DIVEOFF, SORL, and CLUE across various standard D4RL *medium-expert* datasets. Figure 6 shows the diversity of actions selected by each method. Our analysis shows that, compared to other approaches, our intrinsic reward mechanism enables agents to consistently choose from a broader range of actions, fostering greater action diversity.

## 6 CONCLUSION

In this paper, we introduce a Unique Behavior objective function, a novel approach to enhancing diversity in offline RL. Unlike conventional methods that rely on dataset heterogeneity to foster diverse agent behaviors, our approach leverages mutual information to encourage unique and effective strategies across agents—without requiring an additional estimator. Empirical evaluations demonstrate that our framework consistently outperforms existing methods, effectively generating diverse behaviors even in environments with limited data diversity. Our work advances offline RL, extending its applicability to dynamic, unpredictable real-world scenarios.

## REPRODUCIBILITY STATEMENT

To facilitate reproducibility, we provide our full source code and implementation details at https://1drv.ms/u/c/f80c15fe86484a40/EQRuVZ4itwBLn9FdSexvi0MBNeYxcg0bXMK_UBXDxEcveg?e=sgE0ny. All hyperparameter values, training configurations, and evaluation settings are documented in Appendix A.2.

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

# A  APPENDIX

## A.1  LLM USAGE

We utilized Large Language Models (LLMs), specifically ChatGPT, solely for polishing the writing in our paper—such as improving grammar and clarity—and for adding documentation and comments to our codebase. No part of the experimental design, algorithm development, or result analysis was generated or influenced by ChatGPT.

## A.2  HYPERPARAMETER SETTING

For the general hyper-parameters of RL training, we follow EDAC's setting (An et al., 2021) as shown in Table 2. For diversity optimization, the additional hyperparameter $\lambda$, which controls the weight of the uniqueness loss, is set to 1 across all datasets, except in the *medium-expert Walker2d* environment of the diverse D4RL dataset, where it is reduced to 0.5. We train our models on NVIDIA 4090 GPUs.

Table 2: Hyperparameters used in our experiments. *Note:* `exp` = expert, `rep` = replay.

| Hyperparameter / Dataset | Medium Halfcheetah | Medium-Rep Halfcheetah | Medium-Exp Halfcheetah | Medium Hopper | Medium-Rep Hopper | Medium-Exp Hopper | Medium Walker2d | Medium-Rep Walker2d | Medium-Exp Walker2d |
|---|---|---|---|---|---|---|---|---|---|
| actor_learning_rate | 0.0003 | 0.0003 | 0.0003 | 0.0003 | 0.0003 | 0.0003 | 0.0003 | 0.0003 | 0.0003 |
| alpha_learning_rate | 0.0003 | 0.0003 | 0.0003 | 0.0003 | 0.0003 | 0.0003 | 0.0003 | 0.0003 | 0.0003 |
| batch_size | 256 | 256 | 256 | 256 | 256 | 256 | 256 | 256 | 256 |
| buffer_size | 1000000 | 1000000 | 2000000 | 1000000 | 1000000 | 2000000 | 1000000 | 1000000 | 2000000 |
| critic_learning_rate | 0.0003 | 0.0003 | 0.0003 | 0.0003 | 0.0003 | 0.0003 | 0.0003 | 0.0003 | 0.0003 |
| gamma | 0.99 | 0.99 | 0.99 | 0.99 | 0.99 | 0.99 | 0.99 | 0.99 | 0.99 |
| hidden_dim | 256 | 256 | 256 | 256 | 256 | 256 | 256 | 256 | 256 |
| max_action | 1.0 | 1.0 | 1.0 | 1.0 | 1.0 | 1.0 | 1.0 | 1.0 | 1.0 |
| num_critics | 10 | 10 | 10 | 50 | 50 | 50 | 10 | 10 | 10 |
| num_epochs | 3000 | 3000 | 3000 | 3000 | 3000 | 3000 | 3000 | 3000 | 3000 |
| num_updates_on_epoch | 1000 | 1000 | 1000 | 1000 | 1000 | 1000 | 1000 | 1000 | 1000 |
| tau | 0.005 | 0.005 | 0.005 | 0.005 | 0.005 | 0.005 | 0.005 | 0.005 | 0.005 |
| lambda | 1.0 | 1.0 | 1.0 | 1.0 | 1.0 | 1.0 | 1.0 | 1.0 | 0.5 |

## A.3  MOTIVATION AND JUSTIFICATION FOR USING $p^t(s_t \mid \pi^i)$

Under the standard Markov Decision Process (MDP) formulation, the one-step transition probability $P(s_t \mid s_{t-1}, a_{t-1})$ is independent of the agent's identity or policy $\pi$. While this holds in many simplified environments, we explicitly introduce $p^t(s_t \mid \pi^i)$, which denotes the probability of being in state $s_t$ at time $t$ under policy $\pi^i$, to accommodate richer, more realistic scenarios where historical dependencies and agent-specific influences matter.

### A.3.1  MOTIVATION FOR INTRODUCING $p^t(s_t \mid \pi^i)$

Our goal is to define a general formulation that reflects the diversity of agent behavior in complex environments. This includes settings where agents accumulate distinct histories, maintain different belief states, or are subject to context-driven dynamics. By incorporating $\pi^i$ into the state visitation distribution, we account for how an agent's policy can influence its state trajectory, even when underlying dynamics are shared.

### A.3.2   EXAMPLE 1: PARTIALLY OBSERVABLE MDP (POMDP)

In POMDPs, agents do not observe the full state but instead maintain a belief $b_t^i$ over possible states. These beliefs are updated based on observation histories and the agent's action policy. Following Zamboni et al. (2024), the belief trajectory probability includes policy-dependent updates:

$$p^t(s_t \mid \pi^i) = O(o_t \mid s_t) \cdot P(s_t \mid s_{t-1}, a_{t-1}) \cdot T^{o_{t-1}, a_{t-1}}(b_t^i \mid b_{t-1}^i), \tag{7}$$

where $T$ is the belief update operator and $O$ is the observation model. Because the belief update process depends on both the observation sequence and the agent's actions, it varies across agents with different policies.

Due to each agent maintaining its own belief state model (Singh et al., 2021), the variability in these belief states $b_t^{\pi^i}$ leads to diverse values of $p^t(s_t \mid \pi^i)$ among the agents. This makes the inclusion of policy identity $\pi^i$ essential when modeling state visitation distributions in partially observable settings.

### A.3.3   EXAMPLE 2: CONTEXTUAL MDP

Contextual MDPs (Tennenholtz et al., 2023) introduce history-dependent context variables $x_t$ that influence transitions. These contexts evolve based on each agent's past interactions, making them implicitly policy-dependent:

$$p^t(s_t \mid \pi^i) = p(s_t \mid s_{t-1}, a_{t-1}, x_{t-1}^{\pi^i}), \tag{8}$$

where $x_{t-1}^{\pi^i}$ denotes the context experienced by agent $\pi^i$. Different histories yield different context variables and, consequently, different transition behaviors.

### A.3.4   GENERALIZATION ACROSS ENVIRONMENTS

The definition of $p^t(s_t \mid \pi^i)$ is applicable to a wide range of frameworks beyond classical MDPs, including POMDPs, contextual MDPs, and non-Markovian or history-aware systems. This generality enhances the applicability of our diversity formulation in both one-step and $k$-step decision processes, supporting more robust modeling of real-world agent behavior.

### A.3.5   RELATIONSHIP BETWEEN BEHAVIOR AND PATH UNIQUENESS

As discussed in main paper, path uniqueness is defined as:

$$U_{\text{Path}}^{\pi^i}(\tau) = \sum_{t=0}^{T} \log \frac{p(a_t \mid s_t, \pi^i)}{p(a_t \mid s_t)} + \sum_{t=0}^{T} \log \frac{p^t(s_t \mid \pi^i)}{p^t(s_t)}. \tag{9}$$

While the second term is intractable in offline RL, the first term, which defines *behavior uniqueness*, serves as a lower bound. Thus, our Unique Behavior (UB) objective provides a tractable surrogate for path-level diversity that generalizes across standard MDPs, POMDPs, and contextual MDPs.

### A.4   COMPARISON WITH OFFLINE RL METHODS THAT LEARN A SINGLE SOLUTION

We conducted a performance comparison of leading offline reinforcement learning methods, including EDAC (An et al., 2021), Conservative Q-Learning (CQL) (Kumar et al., 2020), and Implicit Q-Learning (IQL) (Kostrikov et al., 2022), using the standard D4RL datasets. Each method was evaluated across five random seeds and ten episodes per seed. Since these methods operate within a single-policy framework, focused solely on performance, no diversity metrics are reported for them in the comparison. Notably, the performance of our diverse solutions is comparable to these single-policy methods. Detailed results from this analysis are presented in Table 3.

Table 3: We present a comparison with leading offline RL baselines, which focus exclusively on performance. Notably, our diverse solution achieves performance comparable to these single-policy methods while also offering the benefit of diverse behaviors.

| Datasets | Metrics | Ours | EDAC | CQL | IQL |
|---|---|---|---|---|---|
| Medium-Expert Hopper | Performance | 95.66±12.04 | **110.7±0.1** | 96.9 ± 15.1 | 85.5 ± 29.7 |
| | State Diversity | **0.76±0.11** | - | - | - |
| | Action Diversity | **0.46±0.22** | - | - | - |
| Medium-Expert Walker2d | Performance | 113.35±0.57 | **114.7±0.9** | 109.1 ± 0.2 | 112.1 ± 0.5 |
| | State Diversity | **0.55±0.04** | - | - | - |
| | Action Diversity | **0.78±0.17** | - | - | - |
| Medium-Expert Halfcheetah | Performance | 95.08±7.88 | **106.3±1.9** | 95.0 ± 1.4 | 92.7 ± 2.8 |
| | State Diversity | **0.82±0.34** | - | - | - |
| | Action Diversity | **0.83±0.16** | - | - | - |
| Medium-Replay Hopper | Performance | **101.41±0.26** | 101.0±0.5 | 86.3 ± 7.3 | 89.6 ± 13.2 |
| | State Diversity | **0.76±0.17** | | | |
| | Action Diversity | **0.42±0.25** | | | |
| Medium-Replay Walker2d | Performance | 79.63±0.54 | **87.1±2.3** | 76.8 ± 10.0 | 75.4 ± 9.3 |
| | State Diversity | **0.85±0.13** | - | - | - |
| | Action Diversity | **0.97±0.03** | - | - | - |
| Medium-Replay Halfcheetah | Performance | **60.92±1.3** | 61.3±1.9 | 45.3 ± 0.3 | 42.1 ± 3.6 |
| | State Diversity | **0.94±0.03** | - | - | - |
| | Action Diversity | **0.98±0.02** | - | - | - |
| Medium Hopper | Performance | **101.44±0.19** | 101.6±0.6 | 61.9 ± 6.4 | 65.2 ± 4.2 |
| | State Diversity | **0.59±0.31** | - | - | - |
| | Action Diversity | **0.48±0.31** | - | - | - |
| Medium Walker2d | Performance | 89.66±0.61 | **92.5±0.8** | 79.5 ± 3.2 | 80.7 ± 3.4 |
| | State Diversity | **0.85±0.15** | - | - | - |
| | Action Diversity | **0.85±0.19** | - | - | - |
| Medium Halfcheetah | Performance | **65.89±0.34** | 65.9±0.6 | 46.9 ± 0.4 | 50.0 ± 0.2 |
| | State Diversity | **0.72±0.21** | - | - | - |
| | Action Diversity | **0.77±0.13** | - | - | - |

## A.5 Algorithm

In Algorithm 1, we implement the uniqueness objective in Eq. (5) by treating it as an intrinsic reward. Concretely, the diversity term is added to the immediate reward in the Bellman target, so that the critic learns a value function for the diversity-augmented return. The policy is then updated purely through this shaped value function, receiving a single, unified value-guided learning signal that already reflects both task performance and behavioral uniqueness.

We also explored an alternative formulation in which the policy optimization combined two separate loss components, one for return and one for diversity. In practice, this caused the policy to receive gradients from two misaligned sources (the value function and the diversity objective), which often led to unstable learning. By instead incorporating the diversity term as intrinsic reward and letting the critic (stabilized by an ensemble) absorb both objectives, we observed noticeably more stable training. This design provides a practical and robust way to optimize the uniqueness objective within a standard Q-learning framework.

## A.6 Additional Visualization

In Figure 7 below we provide an additional visualization demonstrating the effectiveness of our UB objective.

**Algorithm 1** Offline Policy Optimaization with the Unique Behavior objective function

1: **Input:** dataset $D = \{(s_i, a_i, r_i, s_i')\}_{i=1}^{|D|}$
2: **Initialize:** the policies $\pi_\theta^1, ..., \pi_\theta^M$, critics $Q_{w_j}^1, ..., Q_{w_j}^M$ for $j = 1, .., N$
3: **for** $t = 1$ to $T$ **do**
4:      Sample a minibatch $\{(s_i, a_i, s_i', r_i)\}_{i=1}^B$ from $D$
5:      **for** $m = 1$ to $M$ **do**
6:          Compute the target value:

$$y_i^m = r_i + \lambda U_{\text{Behavior}}^{\pi_\theta^m}(a_{i,m}'|s_i') + \gamma \min_{j=1,...,N} Q_{w_j}^m(s_i', a_{i,m}'), \text{ where } a_{i,m}' \sim \pi_\theta^m(\cdot \mid s_i')$$

7:          Update critic: for $j = 1, .., N$,    $B^{-1} \sum_{i=1}^B (y_i^m - Q_{w_j}^m(s_i, a_i))^2$
8:          Update policy:

$$B^{-1} \sum_{i=1}^B (\min_{j=1,...,N} Q_{w_j}^m(s_i, a_{i,m}) - \beta \log \pi_\theta^m(a_{i,m}|s_i)), \text{where } a_{i,m} \sim \pi_\theta^m(\cdot|s_i)$$

9:      **end for**
10: **end for**

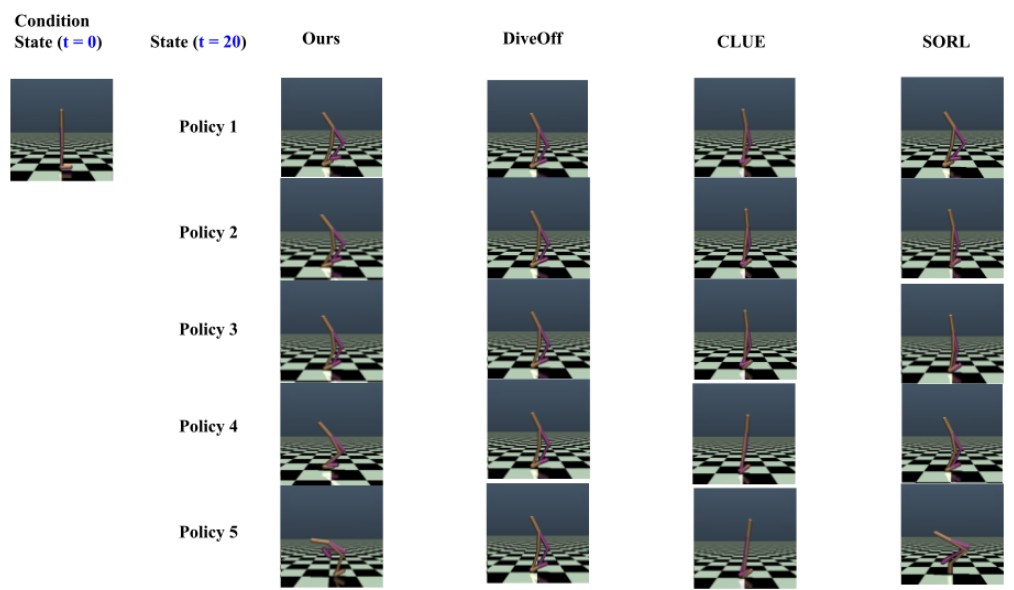

Figure 7: To visually compare the diversity between policies trained with baseline methods and our approach, we initialized the policies from a consistent initial state at $t = 0$ and rendered the resulting states after 20 steps.

## A.7 ABLATION STUDY

We conducted an ablation study across various environments in Mujoco to assess the impact of the proposed the Unique Behavior objective function. The detailed experimental results are provided in Table 4.

## A.8 ABLATION STUDY ON THE NUMBER OF AGENTS

We perform an ablation study on the `halfcheetah-medium-v2` dataset to evaluate how the number of agents (2, 3, or 4) affects the diversity of the learned policies. Diversity is measured according to the metric described in Section 5.1. Higher values indicate greater diversity.

Table 4: Detailed Results of the ablation study. In this table, setting $\lambda = 0$ means that only performance is considered during policy training. As indicated, our unique behavior (UB) objective enhances the behavior diversity of agents without sacrificing their performance.

| Datasets | Metrics | Ours | λ=0 |
|---|---|---|---|
| *Medium-Expert Hopper* | *Performance* | 95.66±12.04 | **98.94±0.53** |
| | *State Diversity* | **0.76±0.11** | 0.31±0.3 |
| | *Action Diversity* | **0.46±0.22** | 0.08±0.09 |
| *Medium-Expert Walker2d* | *Performance* | **113.35±0.57** | **113.6±0.27** |
| | *State Diversity* | **0.55±0.04** | 0.46±0.28 |
| | *Action Diversity* | **0.78±0.17** | 0.48±0.08 |
| *Medium-Expert Halfcheetah* | *Performance* | 95.08±7.88 | **101.32±3.37** |
| | *State Diversity* | **0.82±0.34** | 0.36±0.29 |
| | *Action Diversity* | **0.83±0.16** | 0.37±0.13 |
| **Medium-Expert-Performance** | | 101.36 | **104.62** |
| **Medium-Expert-Diversity** | | **0.7** | 0.34 |
| *Medium-Replay Hopper* | *Performance* | **101.41±0.26** | 100.06±0.71 |
| | *State Diversity* | **0.76±0.17** | 0.07±0.06 |
| | *Action Diversity* | 0.42±0.25 | **0.47±0.28** |
| *Medium-Replay Walker2d* | *Performance* | 79.63±0.54 | **80.29±0.64** |
| | *State Diversity* | **0.85±0.13** | 0.31±0.08 |
| | *Action Diversity* | **0.97±0.03** | 0.44±0.29 |
| *Medium-Replay Halfcheetah* | *Performance* | **60.92±1.3** | 59.79±1.51 |
| | *State Diversity* | **0.94±0.03** | 0.48±0.38 |
| | *Action Diversity* | **0.98±0.02** | 0.66±0.19 |
| **Medium-Replay-Performance** | | **80.65** | 80.05 |
| **Medium-Replay-Diversity** | | **0.82** | 0.41 |
| *Medium Hopper* | *Performance* | **101.44±0.19** | **101.44±0.06** |
| | *State Diversity* | **0.59±0.31** | 0.11±0.04 |
| | *Action Diversity* | **0.48±0.31** | 0.03±0.03 |
| *Medium Walker2d* | *Performance* | 89.66±0.61 | **90.47±0.75** |
| | *State Diversity* | **0.85±0.15** | 0.73±0.13 |
| | *Action Diversity* | **0.85±0.19** | 0.1±0.04 |
| *Medium Halfcheetah* | *Performance* | **65.89±0.34** | 64.78±0.36 |
| | *State Diversity* | **0.72±0.21** | 0.22±0.03 |

Table 5 summarizes the mean and standard deviation of diversity across agents.

Table 5: Ablation on the number of agents for `halfcheetah-medium-v2`.

| # Agents | State Diversity | | Action Diversity | |
|---|---|---|---|---|
| | Mean | Std | Mean | Std |
| 2 | 0.9807 | 0.0141 | 0.9856 | 0.0167 |
| 3 | 0.8922 | 0.0769 | 0.7941 | 0.1469 |
| 4 | 0.5867 | 0.3279 | 0.6041 | 0.3123 |

Across all settings, we find that the UB objective consistently produces meaningful diversity even with a small number of agents. In particular, using only two or three agents already yields high normalized diversity scores with low variance, demonstrating that UB does not rely on a large ensemble to induce differentiated behaviors. This shows that the method is not sensitive to the exact choice of $M$: smaller values of $M$ still provide strong and stable diversity, making UB practical even when computational resources limit the number of agents that can be deployed.

## A.9 ADDITIONAL ABLATION STUDY

We conducted an ablation study to evaluate the impact of the number of critics and the value of $\lambda$ on performance. Specifically, we tested $n_{\text{critics}} \in \{10, 20, 30\}$ and $\lambda \in \{0.0, 0.5, 1.0\}$. Each configuration was run across 3 different random seeds: 10, 11, and 12. In Figure 8 we report the mean and standard deviation of the evaluation reward across training steps. All experiments were performed on the `halfcheetah-medium-v2` environment. Our findings suggest that the performance of our method is relatively stable and robust to the choice of these hyperparameters in this environment.

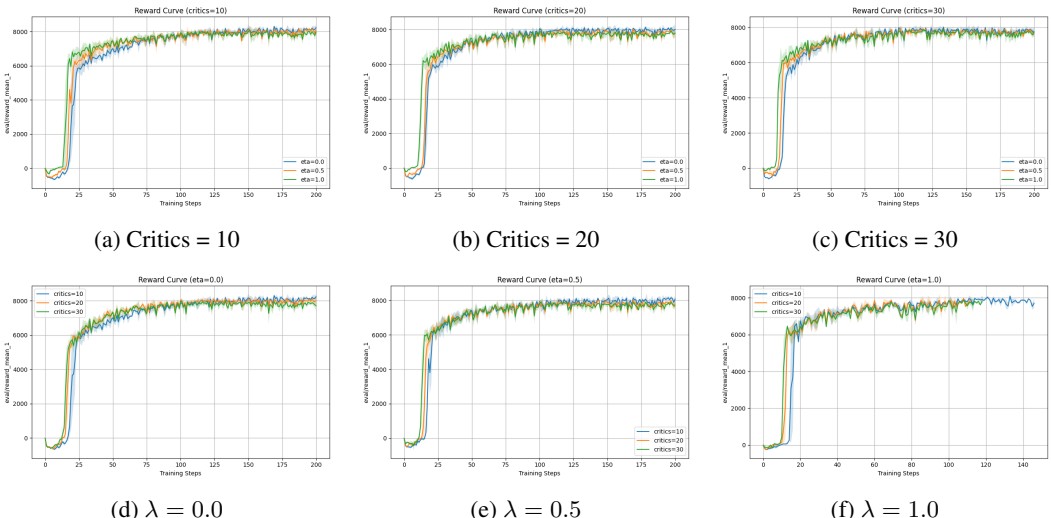

(a) Critics = 10        (b) Critics = 20        (c) Critics = 30

(d) $\lambda = 0.0$        (e) $\lambda = 0.5$        (f) $\lambda = 1.0$

Figure 8: Ablation study results across different numbers of critics and values of $\lambda$ in `halfcheetah-medium-v2`. We plot the evaluation reward mean and standard deviation.

## A.10 CONTROLLABLE DIVERSITY

In Wu et al. (2023), diversity is engineered through the use of user-specified Behavior Descriptors, which promote varying agent behaviors to align with different user-defined criteria. We integrate this concept into our framework for scenarios where specific types of diversity are desirable. By using behavior descriptors, we can tailor the diversity generated by our model to fit particular user needs or ensure alignment with targeted goals.

We conducted experiments on the Maze2d environment as depicted in Figure 3. We define $B(\pi_\theta)$ in Wu et al. (2023) as Figure 9 panel (a), referred to as the user-specified behavior descriptors. Panels (b), (c), and (d) demonstrate the outcomes of this setup. Panel (b) showcases the trajectory of the target agent, which closely follows the target behavior, highlighting the efficacy of our method in embedding and controlling specific agent behaviors. In contrast, Panels (c) and (d) depict the trajectories of other agents who were not given the matching bonus but were still subject to the dataset and unique behavior rewards. These agents exhibit diverse behaviors, diverging significantly from the target, thus emphasizing the diversity achievable under our framework.

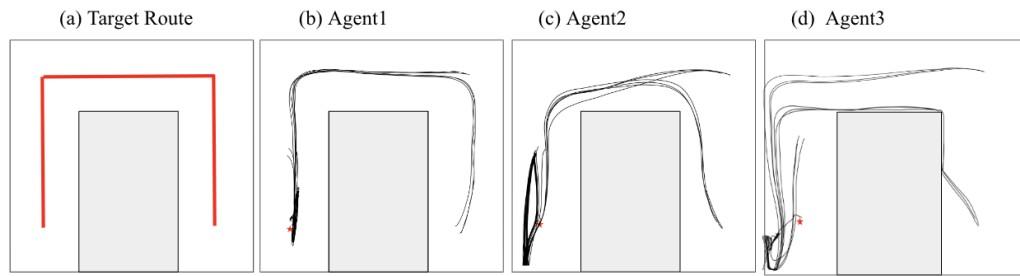

| (a) Target Route | (b) Agent1 | (c) Agent2 | (d) Agent3 |

Figure 9: Controllable Diversity

Table 6: Quality-diversity in discrete action environment

| Atari | Performance | State Diversity | Action Diversity |
|---|---|---|---|
| **SpaceInvaders** | | | |
| Ours | **427.6 ± 50.2** | **0.64 ± 0.29** | **0.56 ± 0.15** |
| SORL | 422.6 ± 84.5 | 0.46 ± 0.25 | 0.27 ± 0.04 |
| **Riverraid** | | | |
| Ours | **1892.8 ± 309.7** | **0.57 ± 0.07** | **0.71 ± 0.16** |
| SORL | 1751.3 ± 313.1 | 0.32 ± 0.15 | 0.50 ± 0.13 |

## A.11 PERFORMANCE IN DISCRETE ACTION ENVIRONMENT

We further extended our evaluation to the Atari domain, where our comparison is primarily with SORL, as it is the only baseline method in our study that has also been tested on Atari environments. We executed the official SORL code available on their GitHub repository to ensure a fair comparison. For this part of our study, we set $\lambda$ to 1. We trained a model with three different policies and tested each across 10 trajectories using three random seeds to assess variability and consistency in performance. The results in Table 6 from these experiments indicate that our method not only performs well across various conditions in MuJoCo tasks but also extends effectively to other environments, including those with discrete action spaces, such as Atari.

