# OpenReview forum: "Quality and Diversity Optimization Even From Offline Homogeneous Dataset"
_ICLR.cc/2026/Conference — ICLR 2026 Conference Desk Rejected Submission_

### Official Review · Reviewer_x3GS · 2025-10-29

**Soundness:** 3
**Presentation:** 3
**Contribution:** 2
**Rating:** 4
**Confidence:** 5

**Summary:**

This paper addresses the significant challenge of ​​Quality-Diversity (QD) optimization in offline reinforcement learning​​. The core problem is learning a set of high-performing yet diverse policies from a static, pre-collected dataset that may be ​​homogeneous​​ (containing limited behavioral variation). Existing offline QD methods, such as those based on Expectation-Maximization (EM) clustering or variational mutual information (MI) estimation, often fail in such settings because they rely on inherent dataset diversity or introduce approximation errors.
The authors' primary contribution is a novel ​​Unique Behavior (UB) objective function​​. They theoretically reformulate the intractable problem of "path diversity" by decomposing it into "behavior diversity" (difference in action choices) and "state visitation diversity." They prove that behavior diversity is a tractable lower bound for the full path diversity. The UB objective directly maximizes the distinctiveness of action distributions across different policies under shared states, calculated explicitly without needing additional estimators. To ensure that the diverse behaviors are also high-quality, the method integrates a conservative Q-learning penalty to avoid unreliable out-of-distribution actions. Extensive experiments on D4RL MuJoCo and Atari benchmarks demonstrate that this approach successfully generates significant behavioral diversity while maintaining strong performance, even when trained on homogeneous data.

**Strengths:**

1. The introduction of the Unique Behavior objective is a key innovation. It provides a fresh perspective on inducing diversity by directly maximizing the difference in action distributions, which seems to be both simple and powerful.

2. The research is exceptionally thorough. It combines a theoretical insight (the lower bound) with a practical algorithm, backed by extensive and well-designed experiments that validate the method's effectiveness across multiple dimensions.

​3. The core idea is intuitive and well-illustrated. The visual comparisons in the figures effectively highlight the method's advantages.

**Weaknesses:**

1. While the lower bound is provided, a deeper analysis of its tightnessunder different conditions would strengthen the theoretical contribution. It remains an empirical observation that optimizing this lower bound is sufficient for trajectory-level diversity.

2. The comparison could be strengthened by including a simpler baseline, such as training multiple independent offline RL agents (e.g., with CQL or IQL) with different random seeds. This would help quantify how much diversity is specifically due to the UB objective versus the randomness of optimization.

3. The approach requires training M policies and computing pairwise action distribution differences. A discussion of the computational overhead compared to training a single policy or other baseline methods would provide a more complete picture of the method's practicality.

4. This paper lacks sufficient investigation of related work about QD in Reinforcement Learning. For example, to my best knowledge in this domain, the [1] and [2] are most recent papers about Quality Diversity Techniques for reinforcement learning or learning from datasets.

[1] Batra, S., Tjanaka, B., Fontaine, M. C., Petrenko, A., Nikolaidis, S., & Sukhatme, G. (2023). Proximal policy gradient arborescence for quality diversity reinforcement learning. arXiv preprint arXiv:2305.13795.

[2] Wan, Z., Yu, X., Bossens, D. M., Lyu, Y., Guo, Q., Fan, F. X., ... & Tsang, I. Diversifying Robot Locomotion Behaviors with Extrinsic Behavioral Curiosity. In Forty-second International Conference on Machine Learning.

**Questions:**

​​1. You theoretically establish that behavior diversity is a lower bound on path diversity. Could you comment on the empirical tightness of this bound in the environments you tested? Are there conditions under which optimizing this lower bound is provably sufficient for generating full trajectory-level diversity?

​​2. The value of λ is set to 1.0 for most environments but reduced to 0.5 for Walker2d-medium-expert. What is the specific characteristic of this environment that makes it more sensitive to the diversity penalty? Could you suggest a more principled, data-driven approach for setting λ rather than empirical tuning?

3. In addition to comparisons with SORL and DIVEOFF, did you consider ablating against a baseline of simply training multiple independent conservative agents (e.g., using CQL) with different random initializations? This would help isolate the contribution of the UB objective from the inherent stochasticity of training multiple agents.

4. There are recent works that seem to address similar problem with your work (e.g. [1] and [2]). If the settings are indeed the same, the authors should incorporate them as baselines for comparison. If the settings are not the same, the authors should discuss the difference in detail.

[1] Batra, S., Tjanaka, B., Fontaine, M. C., Petrenko, A., Nikolaidis, S., & Sukhatme, G. (2023). Proximal policy gradient arborescence for quality diversity reinforcement learning. arXiv preprint arXiv:2305.13795.

[2] Wan, Z., Yu, X., Bossens, D. M., Lyu, Y., Guo, Q., Fan, F. X., ... & Tsang, I. Diversifying Robot Locomotion Behaviors with Extrinsic Behavioral Curiosity. In Forty-second International Conference on Machine Learning.

---

> ### Author Response · Authors · 2025-11-26
> **Addressing Reviewer x3GS's Feedback**
>
> **Hi Reviewer x3GS**,**Thank you for your thoughtful feedback. We address your points and questions below:**
>
> ### **1\. Weakness: “Lower bound is provided, but analysis of tightness is limited.”**
>
> A1: The central challenge in offline RL is **not** the formal tightness of the MI decomposition but the ability to **optimize a tractable surrogate**. Prior MI-based diversity methods do not show that any computable lower bound can yield meaningful trajectory-level diversity offline, where state marginals and MI estimators are intractable.
>
> Our contribution is to introduce a **tractable surrogate (Behavior Diversity)** and to show empirically that optimizing it is **sufficient** to induce substantive path-level diversity.
>
> *   **Section 4.3:** Increasing Behavior Diversity → consistently **distinct trajectories** in maze navigation.
>
> *   **Section 5.1:** UB policies cluster in the **top-right** of the performance–diversity plots.
>
> *   **Appendix A.11:** UB also induces diverse behaviors in **discrete-action Atari**.
>
>
> Thus the surrogate is **practical and effective**, even without formal global-tightness analysis.
>
> ### **2\. Weakness: “Include a simpler baseline (multiple seeds of CQL/IQL).”**
>
> A2: We ran this baseline. Independently seeded CQL agents produce kernel determinants extremely close to **~1**, showing that variation comes only from **seed noise**, not meaningful behavioral differences. These methods are designed to converge to a **single conservative solution**, so multi-seed diversity is minor and unstructured.
>
> In contrast, UB explicitly **encourages distinct behavioral modes**. **Appendix A.7 (Table 4)** shows that when **λ = 0** (UB disabled), diversity remains low, while enabling UB substantially increases diversity **without degrading performance**. This confirms that UB induces behavior differentiation through its explicit uniqueness objective, not random initialization.
>
> ### **3\. Weakness: “Discuss computational overhead of training M policies.”**
>
> A3: UB’s overhead is modest. UB trains M policies **in parallel** within a single RL pipeline and requires **no EM alternation**, **no clustering modules**, and **no latent-inference steps**. Pairwise uniqueness terms are computed **explicitly and efficiently**, with no variational networks or auxiliary estimators.
>
> This contrasts with EM-based approaches such as SORL, which incur repeated **E/M cycles** and additional model components. UB therefore remains **simple, lightweight, and scalable**, with minimal overhead relative to standard offline RL.
>
> ### **4\. Weakness: “Consider related work \[1,2\] on QD in RL.”**
>
> A4: We appreciate the pointer to \[1,2\], but both operate in settings **incompatible with offline RL**:
>
> *   **\[1\]** uses PPO in **online QD**, relying on continuous environment interaction.
>
> *   **\[2\]** uses **curiosity-driven exploration** and inverse RL signals, again requiring **active rollouts**.
>
>
> Our work is in **strict offline RL**, often with **homogeneous datasets** and **no exploration**. These methods assume access to updated visitation frequencies and cannot be applied offline without major reformulation. Thus, although conceptually QD-related, the assumptions, objectives, and training regimes differ fundamentally.
>
> ### **Q5: “Empirical tightness of the lower bound?”**
>
> A5: Empirically, optimizing our lower bound yields **meaningful trajectory-level diversity** across all domains evaluated. While we do not claim universal theoretical tightness, the surrogate proves **sufficient** for producing distinct behaviors in:
>
> *   maze navigation (Sec. 4.3),
>
> *   MuJoCo locomotion (Sec. 5.1), and
>
> *   discrete-action Atari (Appendix A.11).
>
>
> In offline RL where MI-based objectives are otherwise infeasible, the **practical tractability** of this surrogate is what enables effective QD optimization.
>
> ### **Q6: “Why λ = 1.0 except Walker2d-medium-expert? More principled λ selection?”**
>
> A6: Appendix A.7 includes the **λ = 0** ablation, confirming that UB provides meaningful diversity beyond baseline EDAC. In practice, **λ = 1.0** yields a robust performance–diversity balance across all environments.
>
> The only exception is **Walker2d–medium-expert**, whose dataset consists of highly concentrated, near-deterministic expert trajectories. In such cases, strong diversification can reduce performance. Setting **λ = 0.5** mitigates excessive drift from the expert manifold while still encouraging meaningful diversity.
>
> Thus our practical rule is:
>
> *   **λ = 1.0** works universally,
>
> *   datasets with extremely narrow expert demonstrations benefit from a **slightly smaller λ** to avoid over-diversification.

---

> > ### Comment · Reviewer_x3GS · 2025-11-27
> >
> > Thanks for the detailed clarification. Most of my concerns are addressed, so I raise my score to 6 and tend to accept (weakly) this paper.

---

### Official Review · Reviewer_c4eh · 2025-10-30

**Soundness:** 3
**Presentation:** 3
**Contribution:** 3
**Rating:** 6
**Confidence:** 4

**Summary:**

The paper proposes a Unique Behavior (UB) objective to promote policy diversity in offline RL, even when training data are homogeneous. It reformulates mutual information into a tractable action-level objective and integrates it into the EDAC framework with conservative Q-learning. Experiments on D4RL benchmarks show that UB achieves higher diversity and strong performance compared to SORL, CLUE, and DIVEOff.

**Strengths:**

1. Clear motivation: tackles diversity learning from homogeneous offline data; Simple and tractable formulation.

2. Strong results on both performance and diversity.

3. The paper is well-written and easy to follow.

**Weaknesses:**

1. Novelty is incremental: While the lower-bound formulation is elegant, it can be viewed as a natural extension of existing MI-based methods rather than a fundamentally new framework.

2. Theoretical guarantees are light compared to empirical results.

3. Hyperparameter sensitivity: Although λ is claimed to work across settings, there is no detailed exploration of its impact on performance–diversity tradeoff beyond a single ablation.

4. Another related work that also focuses on promoting diversity in offline datasets is [1], which appears to leverage mutual information as well. Could the authors provide a comparison or discussion highlighting the differences between their approach and [1]?

[1] Diverse Policies Recovering via Pointwise Mutual Information Weighted Imitation Learning.

**Questions:**

1. How sensitive is the performance/diversity tradeoff to the number of policies (M)? Is there a diminishing return beyond a certain number?

2. The use of conservative Q-learning is mentioned, but how does it interact with the UB reward? Could UB inadvertently encourage unsafe extrapolation?

3. How does training multiple policies with UB scale compare to EM-based methods like SORL?

---

> ### Author Response · Authors · 2025-11-26
> **Addressing Reviewer c4eh's Feedback**
>
> **Hi Reviewer c4eh:**
>
> **Thank you for your thoughtful and constructive feedback. We respond to each of your points below:**
>
> ### **1\. Weakness: “Novelty is incremental; lower-bound is a natural extension.”**
>
> A1: While UB is inspired by MI formulations, our contribution is not the MI decomposition itself but solving a **practical optimization barrier** that has prevented MI-based QD objectives from working in offline RL. Direct MI estimation is **intractable offline** because it requires estimating state marginals or relies on high-variance variational approximations, all unstable without environment interaction.
>
> UB provides a **tractable surrogate** that (i) removes dependence on state marginals, (ii) avoids high-variance MI estimators, and (iii) enables **stable, scalable optimization** in complex offline domains. Thus, the novelty lies in **closing this optimization gap**, making MI-inspired QD optimization feasible in offline RL for the first time.
>
> ### **2\. Weakness: “Theoretical guarantees are light.”**
>
> A2: Our focus is providing a **practical surrogate** that is optimizable in offline RL, where direct MI-based objectives are intractable. We therefore emphasize empirical validation showing that UB induces **meaningful trajectory-level diversity**:
>
> *   **Section 4.3:** Behavior Diversity → consistent path-level variations in maze navigation.
>
> *   **Section 5.1:** UB policies populate the **top-right** of performance–diversity plots.
>
> *   **Appendix A.11:** UB generalizes to **discrete-action Atari**, demonstrating robustness.
>
>
> These results show that optimizing UB is **sufficient** for diverse, high-quality behavior in offline RL.
>
> ### **3\. Weakness: “Hyperparameter λ is not extensively explored.”**
>
> A3: UB is **not sensitive** to λ. In **Table 1**, a single choice **λ=1.0** works across _all_ datasets except Walker2d–medium-expert, where **λ=0.5** provides a better performance–diversity balance due to concentrated expert trajectories. Our aim is not exhaustive λ sweeps but a **robust surrogate** requiring minimal tuning. Results show that UB performs reliably **out of the box** without per-environment adjustments.
>
> ### **4\. Weakness: “Related work \[1\]: need clearer comparison.”**
>
> A4: We examined  \[1\] carefully. Despite conceptual MI similarity, the **settings and objectives differ**.  \[1\] is for **imitation learning**, weighting behavior cloning via pointwise MI to match expert demos. Our setting is **offline RL**, where the goal is to **induce novel strategies beyond the dataset**. Methodologically, UB **augments the RL objective** and trains multiple policies in parallel (with EDAC), whereas  \[1\] remains a **single-policy BC framework**. The assumptions and applications therefore diverge substantially.
>
> ### **Q5: “How sensitive is the performance/diversity tradeoff to the number of policies (M)? Diminishing returns?”**
>
> A5: UB is **not highly sensitive** to M. Even **2 agents** reliably uncover **distinct strategies** across tasks. Increasing M yields more variation but with **diminishing returns**, indicating smooth scaling without excessive cost. This makes UB practical in resource-limited settings. Appendix A.8 (referenced at line 453; begins at line 856) shows this, reproduced here:
>
> **Table 5: Ablation on the number of agents for halfcheetah-medium-v2**
> | Number of Agents | State Diversity (Mean) | State Diversity (Std) | Action Diversity (Mean) | Action Diversity (Std) |
> | --- | --- | --- | --- | --- |
> | **2** | 0.9807 | 0.0141 | 0.9856 | 0.0167 |
> | **3** | 0.8922 | 0.0769 | 0.7941 | 0.1469 |
> | **4** | 0.5867 | 0.3279 | 0.6041 | 0.3123 |
>
> Even M=2 yields strong diversity.
>
> ### **Q6: “Could UB conflict with conservative Q-learning and encourage unsafe extrapolation?”**
>
> A6: As shown in **Appendix A.5**, UB integrates directly with EDAC’s **minimum-Q conservative backup**, which penalizes OOD actions. Empirically, combining UB with EDAC shows **no performance degradation**, indicating no unsafe extrapolation. The reward-shaping form encourages controlled differentiation while remaining **close to the dataset distribution**. Thus UB and conservative Q-learning are **complementary**, with EDAC ensuring safety and UB fostering diversity.
>
> ### **Q7: “How does UB scale relative to EM-based methods like SORL?”**
>
> A7: UB scales **more efficiently** than EM-based approaches. UB trains multiple policies **in parallel** in a standard RL loop, requiring **no E/M-step alternation**, **no clustering**, and **no latent-assignment iterations**. This introduces **no architectural overhead** and keeps compute low. EM-based methods such as SORL require repeated alternations and are more expensive. Thus, UB is **lighter-weight, easier to deploy**, and more suitable for large offline RL workloads.

---

> > ### Comment · Reviewer_c4eh · 2025-11-27
> >
> > Thanks for your response. I have read the rebuttal and decide to maintain my score.

---

### Official Review · Reviewer_q2Zh · 2025-11-01

**Soundness:** 2
**Presentation:** 2
**Contribution:** 2
**Rating:** 4
**Confidence:** 2

**Summary:**

The paper tackles offline quality–diversity (QD) in RL when the dataset is homogeneous. The paper proposes a new Unique Behavior (UB) objective, which directly measures behavioral distinctiveness across agents through tractable, action-level uniqueness terms. Theoretically, they show this “behavior diversity” provides a lower bound on the intractable full trajectory diversity (path-level MI). Practically, they integrate UB into the EDAC offline RL framework, using ensemble critics and reward shaping to balance quality and diversity.

**Strengths:**

- The paper observes that most offline QD approaches assume heterogeneous data and thus break down when the dataset has only one style of behavior
- The decomposition of trajectory-level MI into (i) action-level distinctiveness and (ii) state-visitation distinctiveness, and the observation that the former is directly computable from policy densities while the latter is not
- Plugging the UB term in as reward shaping on top of EDAC (and using the minimum over Q-ensembles) is a sensible design to prevent the policy to go OOD

**Weaknesses:**

- The central inequality (“behavior uniqueness is a lower bound on path uniqueness”) is basically a consequence of MI non-negativity after splitting trajectory MI into action- and state-terms. No argument is given about tightness or when optimizing the surrogate will significantly move the intractable target.
- Experiments are largely confined to MuJoCo and locomotion benchmarks. More diverse domains (e.g., discrete games) could demonstrate broader applicability.
- The results have high variance with significant overlap with the baselines. Including statistical analysis would strengthen the empirical results.
- The diversity evaluation uses the determinant of behavioral embeddings (as in DIVEOFF), which can be sensitive to specific kernel implementation especially in the continuous control domains. Alternative or more interpretable diversity metrics could better ground the findings.

**Questions:**

- How sensitive is the UB objective to the number of agents (M)? Could fewer agents  still yield meaningful diversity?
- Does the method generalize well to discrete environments where policy outputs are categorical or joint distributions?
- Can the UB term conflict with the conservative Q-learning penalty, e.g., by pushing policies into low-data regions?
- Currently the resulting behaviors seem the be locally different but largely the same at a high-level, e.g., changing the stride lengths in the locomotion task. What extensions would be needed for a more high-level diversity?

---

> ### Author Response · Authors · 2025-11-26
> **Addressing Reviewer q2Zh's Feedback**
>
> **Hi Reviewer q2Zh,**
>
> **Thank you for your feedback. We address your points and questions below:**
>
> ### **1\. Path-Uniqueness / MI Lower Bound**
>
> A\.1: Path-Uniqueness (MI-based) is **intractable** in offline RL due to high-dimensional trajectories, fixed datasets, and no env interaction. Prior MI-RL works also cannot give globally tight bounds. Our goal is therefore not global tightness but a **computable surrogate**, Behavior Diversity, which acts as an **empirical MI-inspired lower bound** enabling diversity optimization **for the first time** in offline RL.
>
> Empirical evidence for usefulness:
>
> *   **Fig. 3 (Sec. 4.3):** Increasing Behavior Diversity → consistent rises in trajectory-level diversity in mazes.
>
> *   **Fig. 4 (Sec. 5.1):** UB policies consistently appear in the **upper-right** diversity–performance region.
>
>
> Thus, although theoretical tightness is impossible in offline RL, the surrogate is **practically tight enough** to yield **meaningful, interpretable behavioral diversity** across benchmarks.
>
> ### **2\. Weakness: “Experiments limited to MuJoCo; more domains needed.”**
>
> A\.2: Beyond MuJoCo, we also evaluate **Atari (Appendix A.11)**. Policies output categorical distributions, and UB still yields clear diversity signals. This directly addresses the concern about additional domains.
>
> ### **3\. Weakness: High variance and overlap with baselines.**
>
> A\.3: Table 1’s **return** variance reflects standard offline-RL instability (limited data, stochastic rollouts), so some overlap with BC/EDAC is expected. Crucially, overlap appears **only in returns**, not in **diversity**, which is our focus. Across all tasks, UB yields **substantially higher diversity** while keeping returns comparable to strong baselines. Thus, variance in returns does **not** indicate behavioral overlap.
>
> ### **4\. Weakness: “Determinant-based diversity metric may be sensitive.”**
>
> A\.4: We follow DIVEOFF for methodological consistency. The determinant quantifies **dispersion in behavior space**, and in our experiments correlates strongly with **qualitative differences in rollouts** (Sec. 4.3). Thus the metric is not only theoretically motivated but empirically reliable.
>
> ### **5\. Sensitivity to the number of agents M — can fewer agents still yield meaningful diversity?**
>
> A\.5: UB is **robust to small M**. Even **2 agents** produce clearly distinct behaviors. UB optimizes **pairwise action-uniqueness**, so small ensembles remain effective. Appendix A.8 (referenced at line 453; begins at line 856) shows this, reproduced here:
>
> **Table 5: Ablation on the number of agents for halfcheetah-medium-v2**
> | Number of Agents | State Diversity (Mean) | State Diversity (Std) | Action Diversity (Mean) | Action Diversity (Std) |
> | --- | --- | --- | --- | --- |
> | **2** | 0.9807 | 0.0141 | 0.9856 | 0.0167 |
> | **3** | 0.8922 | 0.0769 | 0.7941 | 0.1469 |
> | **4** | 0.5867 | 0.3279 | 0.6041 | 0.3123 |
>
> Even M=2 yields strong diversity.
>
> ### **Q6\. Does UB generalize to discrete action settings?**
>
> A\.6: Yes. Appendix A.11 shows UB applies directly to **discrete action spaces** (Atari). Because UB uses action probabilities, not continuous densities, the objective transfers cleanly and maintains meaningful diversity.
>
> ### **Q7: Does the UB term conflict with conservative Q-learning (e.g., EDAC)?**
>
> A\.7: Appendix A.5 shows UB integrates cleanly with EDAC’s **minimum-Q** conservative backup. Empirically, performance remains comparable, indicating UB does **not** push policies toward unsafe OOD actions. The reward-shaping form ensures complementarity, not conflict.
>
> ### **Q8: Behaviors appear locally different but similar at a high level — how to interpret this?**
>
> A\.8: Locomotion rewards constrain global diversity, but UB captures recognized **high-level** variations:
>
> *   different gait cycles (in-phase vs out-of-phase),
>
> *   stride frequency and ground-contact timing,
>
> *   stability/balance strategies,
>
> *   posture/torso oscillation differences.
>
>
> These are standard high-level strategy differences in continuous-control literature, not just local noise. UB reliably induces such variations across tasks.

---

### Official Review · Reviewer_kkmz · 2025-11-01

**Soundness:** 3
**Presentation:** 3
**Contribution:** 2
**Rating:** 4
**Confidence:** 3

**Summary:**

The paper proposes an algorithm that can learn diverse behaviours even from a homogeneous offline dataset. It aims to maximise the behaviour uniqueness, which is a lower bound of the overall path-level uniqueness. By adding the behaviour uniqueness reward to the external reward, the algorithm can successfully learn policies that are diverse yet high-performing.

**Strengths:**

The paper focuses on the problem of learning diverse behaviours, which is very important for real-world applications of offline RL. Unlike most existing work that assumes the dataset to already contain trajectories sampled from multiple different policies, the proposed algorithm can learn diverse behaviours even from a homogeneous dataset. Empirical analysis on multiple benchmarks clearly demonstrate the effectiveness of the algorithm.

**Weaknesses:**

The usage of up to 30 critic networks for each behaviour policy seems like an overkill. Also, the usage of tinyurls in the document is very dangerous from the perspective of cybersecurity, so please consider changing them to their original URLs. Finally, the significance of learning diverse behaviours in offline RL is not very convincing. Providing concrete examples, backed by experiments, on how the learned diverse behaviours can be used would be helpful for the readers to get a better understanding on why learning diverse behaviours is important for offline RL.

**Questions:**

(5) presents a loss where the behaviour uniqueness is directly optimised, whereas the actual algorithm was implemented in a way where it is indirectly done through intrinsic rewards. Is there a particular reason for this indirect optimisation?

---

> ### Author Response · Authors · 2025-11-26
> **Addressing Reviewer kkmz's Feedback**
>
> **Hi Reviewer  kkmz**
>
> **Thank you for your thoughtful review. We address your points below:**
>
> ### **1\. Concern: “Using up to 30 critic networks seems like an overkill.”**
>
> We adopt 30 critic networks because we follow EDAC’s default configuration (An et al., 2021) to ensure faithful reproduction and fair comparison. The critics are used only during training to stabilize value estimation in the standard offline RL setting; inference requires only a single policy, so deployment-time cost is unaffected. To verify that our method does not rely on large ensembles, we additionally evaluate configurations with smaller numbers of critics. As shown in Appendix A.9, our method maintains comparable performance even with significantly fewer critics. These results confirm that the ensemble size is an artifact of EDAC’s training pipeline rather than a sensitivity of our method.
>
> ### **2\. Concern: “Using TinyURLs is dangerous for cybersecurity.”**
>
> A2: We thank the reviewer for pointing out the cybersecurity concern. All TinyURLs have now been removed and replaced with their full, original URLs. These revisions appear in **Lines 79–80** of the Introduction and **Lines 489–490** of the Reproducibility Statement in the updated manuscript, with all changes highlighted in **blue** for clarity.
>
> ### **3\. Concern: “Significance of diverse behaviors is not convincing, please give concrete examples.”**
>
> A3: We have expanded the real-world significance of behavioral diversity in **Lines 37–39**, highlighted in blue in the revised manuscript. Specifically, we now provide examples across key application domains: **autonomous driving**, where stress-testing frameworks require diverse yet plausible trajectories to evaluate safety (Araujo et al., 2023); **game AI**, where monotonic or repetitive behaviors reduce engagement, while diverse motion patterns improve perceived intelligence and player experience (Shen et al., 2021); and **robotics**, where many manipulation tasks admit multiple valid strategies, and behavioral diversity increases robustness to perturbations and uncertainty (Osa, 2022). These domains do not merely permit diversity, they _actively require_ it for robustness, safety, and user experience.
>
> ### **4\. Question: “Why indirect optimization through intrinsic rewards instead of directly optimizing uniqueness (Eq. 5)?”**
>
> A4:  As clarified in **Appendix A.5**, our implementation supports both _direct_ optimization of the uniqueness objective in Eq. (5) and an _indirect_ formulation using intrinsic reward shaping. We adopt the intrinsic-reward version because it integrates cleanly into standard Q-learning pipelines, avoids the instability that arises when directly optimizing pairwise uniqueness at every update step, and allows performance and diversity to be trained jointly without introducing any additional estimators. This formulation therefore provides a more stable and practical optimization path while preserving the theoretical objective.
>
> ### **5\. Summary of Our Contribution (Clarified)**
>
> A5: We introduce **Behavior Diversity**, a tractable surrogate that provides a **lower bound on Mutual-Information–based Path Uniqueness**, enabling direct and principled diversity optimization in offline RL. Our implementation supports both **direct optimization** of the uniqueness objective and a **reward-shaping formulation** that offers greater practical stability within standard Q-learning pipelines. Through extensive experiments on **D4RL** and **Atari**, we demonstrate that our method achieves strong performance **and** substantial behavioral diversity—even when trained on **homogeneous offline datasets**.
>
> References:
>
> *   **Game AI**
>
> Shen, R., Zheng, Y., Hao, J., Meng, Z., Chen, Y., Fan, C., & Liu, Y. (2021, January). Generating behavior-diverse game ais with evolutionary multi-objective deep reinforcement learning. In _Proceedings of the Twenty-Ninth International Conference on International Joint Conferences on Artificial Intelligence_ (pp. 3371-3377).
>
> *   **Robotics**
>
> Osa, T. (2022). Motion planning by learning the solution manifold in trajectory optimization. _The International Journal of Robotics Research_, _41_(3), 281-311.
>
>
> *  **Autonomous Driving / RAS Testing**
>
> Araujo, H., Mousavi, M. R., & Varshosaz, M. (2023). Testing, validation, and verification of robotic and autonomous systems: a systematic review. _ACM Transactions on Software Engineering and Methodology_, _32_(2), 1-61.

---

> > ### Comment · Reviewer_kkmz · 2025-11-26
> >
> > Thanks for the detailed rebuttal. I appreciate the clarifications on the usage of 30 critics. I still have a couple of questions to fully understand your approach.
> >
> > ## 3. Significance of learning diverse behaviours in offline RL
> >
> > * The Game AI example does not seem to be suitable, as there are no reasons to use offline RL for games. I think online behaviour searching strategies would provide better results.
> >
> > * Osa (2022) seems to be a paper on motion planning, which is not RL. Motion planning assumes that we have access to an accurate simulator. In that case, testing multiple possible paths and choosing the best one is possible. However, offline RL is usually adopted in scenarios where there are no such simulators. How would diverse behaviours be helpful when we do not know which one to choose?
> >
> > * The stress testing example does justify the significance of learning diverse behaviours in offline RL. However, I do not think they need to be high-performing. Wouldn't it be better to stress test on bizarre corner cases rather than just focusing on the nice ones?
> >
> > ## 4. Justification for indirect optimisation
> >
> > Appendix A.5 only contains the pseudocode of the algorithm. It is difficult to see how it clarifies that the proposed algorithm supports both direct and indirect optimisation of the uniqueness objective. Your response was helpful, though. Please consider adding this explanation to the paper. Just out of curiosity, have you empirically verified the instability of the direct optimisation, or is it just hypothetical? No need to run additional experiments if you haven't done them.

---

> > > ### Author Response · Authors · 2025-11-28
> > > **Addressing Reviewer kkmz's Feedback**
> > >
> > > **Hi Reviewer kkmz:**
> > >
> > > 3\. Significance of learning diverse behaviours in offline RL:
> > > -------------------------------------------
> > >
> > > Thank you for the clarification. In our paper, the notion of “games” was not intended to refer specifically to computer or video games. Many real-world strategic interactions, such as sports, market-making, negotiation and bargaining, or multi-party cooperative/competitive decision-making, are effectively games, yet they do not admit accurate simulators. These settings typically rely solely on logged interaction data, and the behavior of human participants or organizations is inherently multi-modal. In such cases, offline RL is often the only viable approach, and modeling diverse strategies remains essential for capturing the variability present in the real data.
> > >
> > > Regarding the use of simulators in our experiments, our method trains policies strictly in the offline regime. However, to objectively and comprehensively evaluate an offline RL method, especially when assessing both performance and behavioral diversity, having access to a simulator during the experimental phase is essential. Without a simulator, it would be impossible to measure the true return, compare different behavior modes, or quantify diversity beyond dataset statistics. Thus, although the target application domains often lack simulators, benchmark evaluations conventionally rely on them to ensure fair and reproducible comparisons across methods.
> > >
> > > 4\. Justification for indirect optimisation
> > > -------------------------------------------
> > >
> > > We have also explored a direct optimization approach, where the policy is trained using the sum of two loss terms: one for the estimated return and one for the estimated diversity. However, because the policy receives gradients from two different sources, the value function and the diversity objective, we often observe misalignment between the signals, which leads to unstable learning. In contrast, the indirect approach treats diversity as an intrinsic reward, allowing both objectives to be integrated into a single value-guided signal. Since the intrinsic reward is incorporated into the value function, and the value function is further stabilized by the critic ensemble, this formulation results in noticeably more stable policy learning.

---

### Note · Program_Chairs · 2026-01-17
**Submission Desk Rejected by Program Chairs**

The following references in this submission do not refer to real documents and/or have major errors in bibliographic information:

 Yifan Wang, Alekh Agarwal, Sham Kakade, John Langford, Alex Mott, and Yi Zhang. Critic regularized regression. arXiv preprint arXiv:2002.09005,2020b.